# LLM4FL: Multi-Agent Repository-Level Software Fault Localization via Graph-Based Retrieval and Iterative Refinement

## Abstract

Locating and fixing software faults is a time-consuming and resource-intensive task in software development. Traditional fault localization methods, such as Spectrum-Based Fault Localization (SBFL), rely on statistical analysis of test coverage data but often lack accuracy. While more effective, learning-based techniques require large training datasets and can be computationally intensive. Recent advancements in Large Language Models (LLMs) have shown potential for improving fault localization by enhancing code comprehension and reasoning. LLMs are typically pretrained and can be leveraged for fault localization without additional training. However, these LLM-based techniques face challenges, including token limitations, performance degradation with long inputs, and difficulties managing large-scale projects with complex, interacting components. We introduce LLM4FL, a multi-LLM-agent-based fault localization approach to address these challenges. LLM4FL utilizes three agents. First, the Context Extraction Agent uses an order-aware division strategy to divide and analyze extensive coverage data into small groups within the LLM's token limit, identify the failure reason, and prioritize failure-related methods. The prioritized methods are sent to the Debugger Agent, which uses graph-based retrieval to identify failure reasons and rank suspicious methods in the codebase. Then the Reviewer Agent re-evaluates and re-ranks buggy methods using verbal reinforcement learning and self-criticism. Evaluated on the Defects4J (V2.0.0) benchmark of 675 faults from 14 Java projects, LLM4FL outperforms AutoFL by 18.55% in Top-1 accuracy and surpasses supervised methods like DeepFL and Grace, all without task-specific training. Coverage splitting and prompt chaining further improve performance, boosting Top-1 accuracy by up to 22%.

## 1 Introduction

The process of locating and fixing software faults requires significant time and effort. Research shows that software development teams allocate more than half of their budgets to testing and debugging activities (Hait & Tassey, 2002; Alaboudi & LaToza, 2021). As software systems become increasingly complex, the demand for more accurate fault localization techniques grows. To assist developers and reduce debugging costs, researchers have developed various fault localization techniques (Li et al., 2019; Lou et al., 2021; Qian et al., 2023b; Li et al., 2021; Abreu et al., 2009; Sohn & Yoo, 2017). These techniques analyze code coverage and program execution to identify the most likely faulty code, assisting developers in finding the fault.

Despite the advances in fault localization, many existing techniques still struggle with scalability and precision. Traditional methods, such as Spectrum-Based Fault Localization (SBFL), use statistical analysis to analyze coverage data from passing and failing test cases to rank suspicious code elements (Abreu et al., 2006). While these techniques provide valuable insights, their accuracy is lower. Their reliance on statistical correlations between test failures and code coverage does not always capture the deeper semantic relationships needed for more accurate fault localization (Wong et al., 2016; Xie et al., 2013; Le et al., 2013). Recent techniques applied machine learning (Sohn & Yoo, 2017; Zhang et al., 2019a; Li et al., 2021; 2019) and deep learning models (Lou et al., 2021; Qian et al., 2023b; Rafi et al., 2024; Lou et al., 2021) to address these

issues to improve fault localization. These methods enhance the ranking of suspicious code elements by incorporating additional information like code complexity, text similarity, and historical fault data. However, these techniques often require extensive training data that may not be available.

Recent advances in Large Language Models (LLMs) have shown great potential for software fault localization due to their strong language comprehension and generation capabilities (Abedu et al., 2024; Lin et al., 2024a). LLMs trained on extensive programming datasets can understand code structure, interpret error messages, and even suggest bug fixes (Wu et al., 2023; Pu et al., 2023; Li et al., 2023). These models, with their ability to analyze and process both natural language and code, present an opportunity to significantly improve traditional fault localization methods by incorporating deeper semantic analysis and context-aware reasoning.

Several recent studies have explored the use of large language models (LLMs) for fault localization (Kang et al., 2024; Qin et al., 2024; Zhang et al., 2024), showing promising results. However, these approaches still have important limitations. As summarized in Table 1, current techniques face three key challenges: ① Due to LLM's token limitation, prior studies often use only method names or summaries to determine if a method requires further investigation, thereby missing code implementation details. ② They rely on text-based search to navigate across methods and files based on names, making them prone to LLM hallucinations and confusion caused by methods having similar names or overridden methods. ③ They make one-shot decisions when ranking faulty methods, limiting their ability to improve the ranking when there is new information.

In this paper, we propose *LLM4FL*, a multi-agent LLM-based fault localization technique designed to analyze software projects at the repository level, emulating developers' fault localization process. To address ①, *LLM4FL* uses the *Context Extraction Agent*, which mitigates token constraints by dividing covered methods into manageable groups and analyzing every method's implementation details to prioritize the most suspicious ones. This allows the LLM to reason over source code rather than rely solely on method names or summaries. To address ②, the *Debugger Agent* enhances structural awareness by performing graph-based, retrieval-augmented code navigation. It follows interprocedural call relationships when navigating the codebase, ensuring that retrieved methods are semantically and structurally relevant, rather than based solely on textual similarity. Finally, to address ③, the *Reviewer Agent* refines the results through verbal reinforcement learning (Shinn et al., 2024), iteratively revisiting and improving the ranking of suspicious methods based on earlier reasoning and method implementation details. This iterative refinement leads to more stable and accurate fault localization outcomes.

We evaluated *LLM4FL* on the Defects4J (v2.0.0) benchmark (Just et al., 2014), which includes 675 real-world faults across 14 open-source Java projects. As a widely adopted dataset in fault localization research (Kang et al., 2024; Qin et al., 2024; Lou et al., 2021; Rafi et al., 2024; Li et al., 2019), Defects4J provides a controlled environment for evaluating FL techniques. A recent study (**?**) also shows that it has a very low risk of data leakage for LLMs, making it suitable for evaluating LLM-based FL techniques. Our results demonstrate that **LLM4FL** surpasses LLM-based technique **AutoFL** (Kang et al., 2024) and **AgentFL/SoapFL** (Qin et al., 2024) by achieving 18.55% and 4.82% higher Top-1 accuracy, respectively. *LLM4FL* is also the cheapest among the three, costing only $0.05 per fault. Additionally, *LLM4FL* outperforms supervised techniques such as *DeepFL* (Li et al., 2019) and *Grace* (Lou et al., 2021), even without task-specific training. We also analyzed the impact of individual components within *LLM4FL* on fault localization accuracy. Our findings indicate that each component plays a significant role in performance. Among these, dividing the covered methods into smaller groups and Graph-RAG-based code navigation contribute the most.

Moreover, we examined whether the initial ordering of methods provided to the LLM influences performance. The results reveal that method ordering is important. Even though LLM eventually visits all methods, the Top-1 accuracy still has a difference of up to 22% when comparing an execution-based ordering and the order provided by *DepGraph* (Rafi et al., 2024). We also find that by combining traditional techniques with *LLM4FL*, we can further improve the fault localization accuracy.

The paper makes the following contributions:

- We introduce *LLM4FL*, a novel LLM-based fault localization technique that employs a divide-and-conquer strategy. This technique groups large coverage data and ranks the covered methods using an

SBFL formula. Using multiple agents and code navigation based on Graph-RAG, *LLM4FL* analyzes the repository iteratively to identify and localize faults.

- We conducted an extensive evaluation and *LLM4FL* demonstrates superior performance, surpassing *AutoFL* (Kang et al., 2024) by 18.55% and *AgentFL/SoapFL* (Qin et al., 2024) by 4.82% in Top-1 accuracy. It also outperforms supervised techniques like *DeepFL* and *Grace*, achieving these results without requiring task-specific training.

- We find that the different ordering on the initial method list passed to LLM can affect fault localization accuracy by up to 22% in Top-1 scores, even though LLM eventually visits all the methods.

- Our analysis of *LLM4FL*'s components shows that key components like dividing methods into smaller groups and code navigation are essential to its fault localization accuracy. Removing these features leads to performance declines, emphasizing their importance in handling token limitations and effective fault analysis.

- The data and source code of this work are publicly available online: `https://anonymous.4open.science/r/llm4fl-10AD/readme.md`.

In short, we provide a strategy to mitigate the token limitation issues and analyze repository-level data in LLM-based fault localization. We also highlight the impact of initial method ordering for LLM's input. The findings may help inspire future research on LLM-based fault localization for large-scale software projects.

**Paper Organization.** Section 2 discusses background and related work. Section 3 describes our technique, *LLM4FL*. Section 4 presents the experimental results. Section 5 discusses the threats to validity. Section 6 concludes the paper.

## 2 Related Work

**Large Language Models.** Large Language Models (LLMs), primarily built on the transformer architecture (Meta AI, 2024; Brown, 2020; Roziere et al., 2023), have significantly advanced the field of natural language processing (NLP). These LLMs, such as the widely recognized GPT3 model with 175 billion parameters (Brown, 2020), are trained on diverse text data from various sources, including source code. The training involves self-supervised learning objectives that enable these models to develop a deep understanding of language and generate contextually relevant and semantically coherent text. LLMs have shown substantial capability in tasks that involve complex language comprehension and generation (Abedu et al., 2024; Lin et al., 2024a), such as code recognition and generation. Recent research has leveraged LLMs in software engineering tasks, particularly in fault localization (Kang et al., 2024; Qin et al., 2024; Yang et al., 2024), where they assist in identifying the faulty code groups responsible for software errors. One of the key advantages of using LLMs in fault localization is their ability to process both natural language and code without re-training, allowing them to analyze error messages, stack traces, and test case information to infer suspicious methods or code sections in an unsupervised zero-shot setting.

**LLM Agents.** LLM agents leverage LLMs to autonomously execute tasks described in natural language autonomously, making them versatile tools across various domains. LLM agents are artificial intelligence systems that utilize LLMs as their core computational engines to understand questions and generate human-like responses. They leverage functionalities like memory management (Zhou et al., 2023) and tool integration (Xia et al., 2024; Roy et al., 2024) to handle multi-step and complex operations seamlessly. The agents can refine their responses based on feedback, learn from new information, and even interact with other AI agents to collaboratively solve complex tasks (Hong et al., 2024; Qian et al., 2023a; Xu et al., 2023; Lin et al., 2024a). Through prompting, agents can be assigned different roles (e.g., a developer or a tester), providing more domain-specific responses that help improve the answer (Hong et al., 2024; White et al., 2024; Shao et al., 2023).

Recent studies (Shinn et al., 2024; Renze & Guven, 2024; Pan et al., 2025) explore using the agent's verbal reasoning result to guide iterative improvement (i.e., **verbal reinforcement learning**), which has shown promising improvement in downstream tasks. In verbal reinforcement learning, LLM agents receive natural

language feedback, such as reasoning results or instructions, from other agents as a reward signal. This allows agents to learn and adapt their behavior based on human-like guidance, improving their learning process to solve complex tasks. As the capabilities of large language model (LLM) agents grow, they play an essential role in enhancing automation and increasing productivity in software development. They can assist in code generation (Nijkamp et al., 2022; Lin et al., 2024b; Gu, 2023), debugging (Lee et al., 2024; Kang et al., 2024), test case creation (Huang et al., 2024; Chen et al., 2024b), and automated refactoring (Pomian et al., 2024; Liu et al., 2025), enabling developers to streamline repetitive tasks and focus on higher-level design and problem-solving. Additionally, LLM agents can facilitate collaborative software engineering by acting as intelligent assistants in code reviews, documentation generation, and issue resolution, improving overall development efficiency (Lin et al., 2024b; He et al., 2024). This paper explores using LLM agents to improve fault localization by emulating developers' debugging process using verbal reinforcement learning.

## 2.1 Related Work

**Spectrum-based Fault Localization.** Spectrum-based fault localization (SBFL) (Abreu et al., 2006; Jones et al., 2002; Wong et al., 2013; Abreu et al., 2009) employs statistical techniques to evaluate the suspiciousness of individual code elements, such as methods, by analyzing test outcomes and execution traces. The core idea of SBFL is that code components executed more frequently in failing tests and less frequently in passing tests are more likely to contain faults. Despite its widespread study, SBFL's practical effectiveness remains limited (Kochhar et al., 2016; Xie et al., 2016). To enhance SBFL's accuracy, recent research (Cui et al., 2020; Wen et al., 2019; Chen et al., 2022; Xu et al., 2020b) has suggested incorporating additional data, such as code changes (Wen et al., 2019; Chen et al., 2022) or mutation analysis (Cui et al., 2020; Xu et al., 2020b). However, SBFL's reliance on code coverage metrics still poses challenges, as its suspiciousness scores may not generalize effectively to different faults or systems.

**Learning-based fault localization.** Recent efforts have focused on improving SBFL with learning-based methods (Sohn & Yoo, 2017; Zhang et al., 2019a; Li et al., 2021; Li & Zhang, 2017; Li et al., 2019; Zhang et al., 2019b). These approaches use machine learning models like radial basis function networks (Wong et al., 2011), back-propagation networks (Wong & Qi, 2009), and convolutional neural networks (Zhang et al., 2019b; Li et al., 2021; Albawi et al., 2017) to estimate suspiciousness scores based on historical faults. Some techniques, such as *FLUCCS* (Sohn & Yoo, 2017), combine SBFL scores with metrics like code complexity, while others, like *DeepFL* (Li et al., 2019) and *CombineFL* (Zou et al., 2019), merge multiple sources such as spectrum-based and mutation-based data (Moon et al., 2014; Papadakis & Le Traon, 2015; Dutta & Godboley, 2021). Graph neural networks (GNNs) have also been applied to fault localization (Qian et al., 2023b; Lou et al., 2021; Qian et al., 2021; Xu et al., 2020a). Techniques like *Grace* (Lou et al., 2021) and *GNET4FL* (Qian et al., 2023b) utilize test coverage and source code structure for improved accuracy, while *DepGraph* (Rafi et al., 2024) refines these approaches by graph pruning and incorporating code change information, resulting in higher performance with reduced computational demands. Although these learning-based techniques show improved results, they require training data that may not be available to every project.

**LLM-Based Fault Localization.** Large Language Models (LLMs), such as GPT-4o (OpenAI, 2024) and LLaMA (Meta AI, 2024), demonstrated remarkable abilities in processing both natural and programming languages. LLMs have shown potential in identifying and fixing errors using program code and error logs (Achiam et al., 2023). Due to LLM's token limitations, some LLM-based fault localization techniques operate on small code snippets. *LLMAO* (Yang et al., 2024) uses bidirectional adapters to score suspicious lines within a 128-line context. In contrast, Wu et al. (Wu et al., 2023) prompt ChatGPT with code and error logs but struggle to scale to large projects (Liu et al., 2024).

Recent LLM-based techniques address these challenges with strategies for retrieving failure-related classes or methods, navigating code repositories, and ranking suspicious methods (Kang et al., 2024; Qin et al., 2024; Zhang et al., 2024). Table 1 summarizes the key differences in how these components are designed across prior work and our approach.

*AutoFL* (Kang et al., 2024) retrieves class and method signatures related to the test failure, and prompts the LLM to decide which methods need future analysis (i.e., to examine the code). While this approach

Table 1: Comparison of LLM-based fault localization techniques.

| Aspect | AutoFL (Kang et al., 2024) | SoapFL/AgentFL (Qin et al., 2024) | AutoCodeRover (Zhang et al., 2024) | LLM4FL (this work) |
|---|---|---|---|---|
| ① **Handling token limitation** | Provide only class and method signatures to the LLM, then let it decide which methods need deeper analysis. | Use LLMs to generate descriptions for classes and methods using their signatures. Then, the LLM decides which classes/methods are relevant to the test failures for further analysis. | Iteratively retrieve classes and methods using text-based search guided by the issue description and prioritize candidates based on SBFL rankings. | Divides methods into groups based on the token size and analyzes each group separately. |
| **Limitations of prior token limit handling techniques** | – Uses only method names or LLM-generated summaries to decide which methods need further analysis: Did not provide implementation details to LLMs, so the model **selects methods to investigation based only on names/descriptions** rather than actual code semantic. | | | LLM4FL analyzes the **full source code** with their interprocedural code structure. |
| ② **Navigating large code repository** | Retrieves class and method signatures related to the test failure, then let the LLM decide which methods need deeper analysis. | Ranks classes based on test-failure data and their documentation, then analyzes each class's methods iteratively, guided by the documentation. | Start with the file or method names extracted from issue reports, then iteratively search the repository based on called methods in the retrieved source code. | The agent autonomously navigates the repository, iteratively following an inter-procedural call graph to retrieve and explore connected methods. |
| **Limitations of prior navigation techniques** | – Lacks awareness of repository-level structure: **May overlook cross-file or inter-method dependencies**, leading to missed or incorrect analysis.
– Relies on text-based code retrieval: **Prone to LLM hallucinations and confusion**, particularly with overridden, non-existent methods, or methods with similar names. | | | LLM4FL **uses interprocedural call graph to guide navigation**, ensuring the retrieved code is correct and reachable. |
| ③ **Ranking of faulty methods** | Re-run the fault localization process multiple times and do a majority vote on the result. | Rank methods by having two LLM agents discuss the code and decide which ones are most likely to be faulty. | Does not rank methods; it selects a method as faulty from the retrieved context and generates a patch. | Hierarchical ranking. First, it finds the suspicious methods and ranks them in each group. Then, it employs verbal reinforcement learning to combine and re-rank the suspicious methods from all groups. |
| **Limitations of existing faulty-method ranking techniques** | – No iterative refinement or revisiting of prior decisions: Rely on **one-shot decision, limiting their ability to improve rankings** based on new information. | | | LLM4FL refines its initial rankings by **revisiting the code and incorporating new insights** through iterative reasoning. |

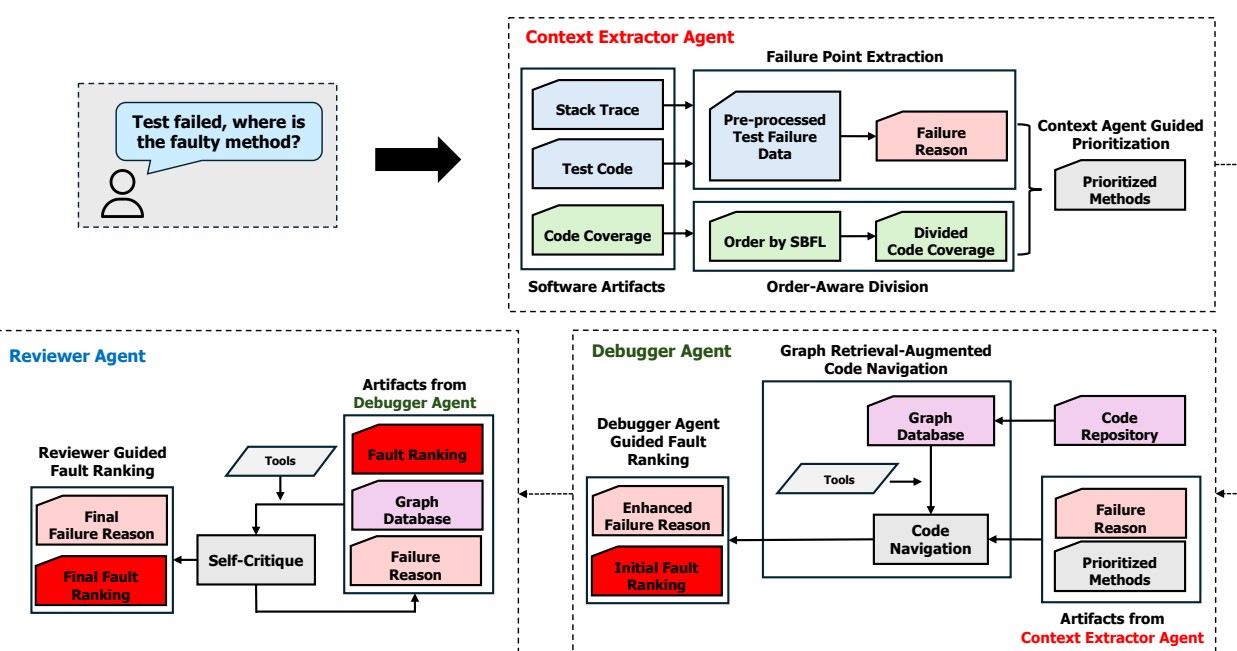

Figure 1: Overview of *LLM4FL* for Multi-Agent Fault Localization, illustrating how agents collaborate to analyze software artifacts, extract failure reasoning, perform graph-based retrieval-augmented code navigation, and rank faulty methods using verbal reinforcement learning.

ensures the input size is less than the LLM's token limitation, it may miss method dependencies in the analysis. Instead, it refines rankings by repeatedly running the process and using a majority voting mechanism to rank faulty methods. **AgentFL/SoapFL** (Qin et al., 2024) models fault localization as a structured operating procedure with controlled phases. It uses LLM-generated summarization of methods/classes or developer-provided documentation to guide code navigation and iteratively scores and ranks suspicious methods to narrow down fault candidates. However, summaries may miss important implementation logics, inter-procedural connections, or bug-related details in the complete code. **AutoCodeRover** (Zhang et al., 2024) extracts method names from issue reports and iteratively searches the codebase for related methods. It may use SBFL to guide this process, but does not rank methods; instead, it selects one from context to generate a patch.

In contrast to prior techniques, *LLM4FL* uses a structured, graph-based approach to analyze the code, enabling the LLM to systematically navigate the repository by considering caller-callee relationships. This reduces LLM hallucination when searching and retrieving methods in the repository. Additionally, *LLM4FL* uses a divide-and-conquer strategy to mitigate token limitations by breaking down large methods into smaller, manageable groups, ensuring comprehensive analysis. Finally, *LLM4FL* refines fault localization results through verbal reinforcement learning, iteratively re-ranking suspicious methods for improved accuracy. These contributions collectively enhance the accuracy and scalability of LLM-based fault localization.

## 3    Methodology

Figure 1 provides an overview of *LLM4FL*. It consists of three LLM-Agents by using novel prompting techniques: (i) *Context Extraction Agent*, (ii) *Debugger Agent*, and (iii) *Reviewer Agent* to localize the fault iteratively. The *Context Extraction Agent* utilizes an **order-aware division** and **failure-reason guided prioritization** prompting technique, which consists of two phases. In the *division phase*, a toolchain divides the large-scale code coverage data into small groups to fit within the LLM's token limits. In the *prioritization phase*, the agent iteratively analyzes the divided code coverage, leveraging failed test cases and stack traces to identify potentially faulty methods. The *Debugger Agent* then performs **graph-based**

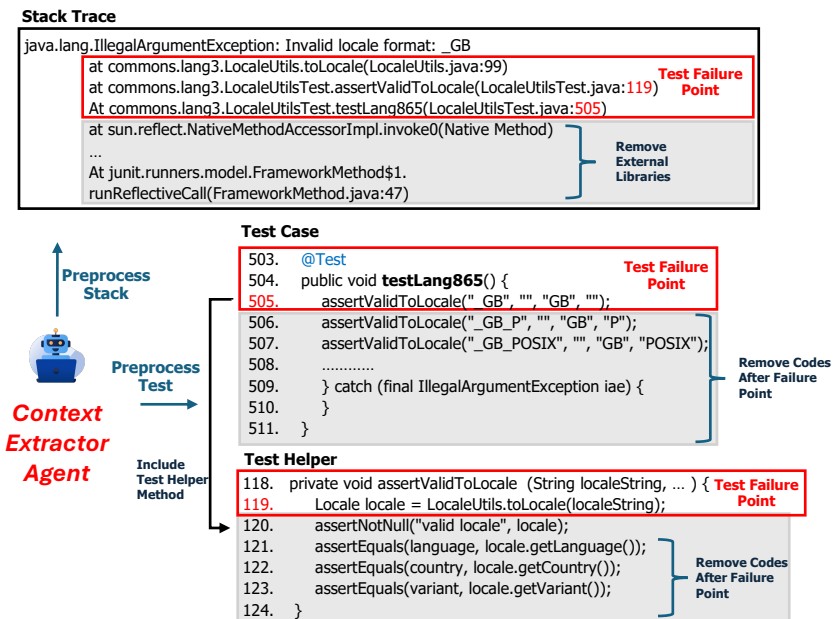

Figure 2: Context-Extraction Agent uses tool-chains to preprocess software artifacts for Lang-5 to emphasize the test failure context. For (i) stack-trace the agent prunes external libraries, and (ii) test code, the agent prunes statements in the test code after the assertion failure.

**retrieval-augmented code navigation** to locate code artifacts further to enhance the accuracy of fault ranking. Finally, the *Reviewer Agent* re-ranks the buggy method through **verbal reinforcement learning**, ensuring more precise fault localization.

## 3.1 Context Extraction Agent

This agent defines a **order-aware division** and **failure-reason guided prioritization** prompting technique to divide code-coverage and iteratively prioritize faulty methods using stack trace and test code within each division. We describe the technique below.

### 3.1.1 Order-Aware Division Phase

To mitigate token size limitation, our *LLM4FL* first performs a division phase, where it first runs the test case to extract the list of method-level code coverage using GZoltar (Campos et al., 2012), which is denoted as $C$, containing a sequence of pairs $(m, s)$, where $m$ denotes the method and $s$ denotes the set of statements within method $m$. We then divide $C$ into $K$ group of sequences of $C_1, C_2, \ldots, C_k$, where $K = \left\lceil \frac{\text{Input Token Length}}{\text{Token Limitation}} \right\rceil$, and each subset satisfies $|C_i| \leq \lfloor \text{Token Limitation} \rfloor$ to ensure the data fits within the LLM's context window. If including $m$ would exceed the limit, it is deferred entirely to the next subset $C_{i+1}$. For example, given an *Input Token Length* of 500K tokens, we divide this by the *Token Limitation* for GPT-4o-mini, which has a limit of 128K tokens (OpenAI, 2024). This results in $K = 4$, each with a 128K token limit. To ensure methods are not split across subsets, we include a method $m$ in subset $C_i$ only if its full token representation fits within the remaining token budget.

Traditionally, the order in which divisions occur in divide-and-conquer algorithms does not impact the outcome. However, prior studies have shown that LLM may improve performance when the order of instructions is carefully considered (Chen et al., 2024a). Inspired by this, we propose *order-aware division*, where we use a Spectrum-Based Fault Localization (SBFL) technique to sort $C$ based on their likelihood of being faulty. Specifically, we use method-level *Ochiai* ranking to order $C_1, C_2, \ldots, C_k$, which is an efficient and unsupervised technique that assigns higher suspiciousness scores to methods that are executed more frequently by

failing test cases and less frequently by passing ones (Abreu et al., 2006; Lou et al., 2021; Li et al., 2021; Cui et al., 2020; Wen et al., 2019; Qian et al., 2021).

### 3.1.2 Failure-Reason Guided Prioritization Phase

To prioritize faulty methods from the divided code coverage, one approach is to provide the agent with the stack trace and test case. However, prior studies have shown that incorporating a summarized description can improve accuracy (Stiennon et al., 2020; Roit et al., 2023). Thus, we propose failure-reason-guided fault prioritization, which follows two steps. First, the input tokens are summarized into a failure-reason representation, capturing the test purpose failure reason. Second, the divided code coverage is prioritized based on the reason for failure. Below, we discuss each step in detail.

**Generate failure-reason from code artifacts.** Our failure-reason generation makes the following three observations: (i) Firstly, the stack traces are verbose and include calls to external libraries unrelated to the fault. (ii) Secondly, some statements within a test case are irrelevant to the failure, specifically statements after the first assertion failures. (iii) Finally, the test case may call other helper methods that trigger the test failure (Peng et al., 2022). Hence, as shown in Figure 2, to mitigate (i), the *Context Extraction Agent* uses `PreprocessTrace` tool-chain to prune external execution in the stack trace. To mitigate (ii) and (iii), *Context Extraction Agent* uses `PreprocessTest` tool-chain, which uses static analysis to build an interprocedural call graph to extract all helper test methods called by the test and prunes all the statements appearing in the test after the failure point. Given the preprocessed stack trace and test code, *Context-Extraction Agents* generates *failure-reason* by generating the test purpose and failure reason. A simplified example of a *failure-reason* generated for *Lang-5* given stack trace and test code is:

> ## **Test Purpose**:
> Test whether LocaleUtils.toLocale parses '_GB' as a locale string, expecting it to return a Locale object with an empty language and 'GB' as the country.
> ## **Failure Reason**:
> The actual output is an IllegalArgumentException due to an invalid locale format, as LocaleUtils.toLocale requires a language before the underscore. For example, '_GB' does not conform to the expected locale format. Update the test input to 'en_GB' or modify LocaleUtils.toLocale to handle '_GB' as a valid case.

**Failure-reason guided prioritization.** Given *Failure-reason*, the *Context Extraction Agent* analyzes the covered methods in each group, previously denoted as $\{C_1, C_2, \ldots, C_K\}$, to prioritize methods that are most related to the failure. Formally, prioritizing each set $C_i$ results in a prioritized subset $C_i'$, where: $C_i' \subseteq C_i, \forall i \in \{1, 2, \ldots, K\}$. The prioritized methods from all subsets are then combined into a final prioritized set: $C' = \bigcup_{i=1}^{K} C_i'$. This final prioritized list $C'$ is just a union of the ordered subsets, meaning no additional processing is introduced. This union process is necessary as LLMs are highly sensitive to initial order (Chen et al., 2024a), yet analyzing entire code coverage poses a token limitation. By dividing coverage into groups, prioritizing each, and unionizing subsets, we extract important contexts that are necessary for the *Debugger Agent* for fault analysis and localization in the next step.

### 3.2 Debugger Agent

Previously, the *Context Extraction Agent* prioritized $C$ to subsets $C'$ to mitigate the LLM's token limitation when analyzing large coverage data. While this technique can adapt to any LLM and token input size, prioritization may miss methods in the coverage that are important for fault localization. Inspired by how developers analyze the call dependencies to understand program execution and identify faults, we propose a **graph-based retrieval-augmented code navigation strategy (Graph-RAG)**. This approach enables the *Debugger Agent* to utilize call graphs and method bodies for effective repository navigation, implementation analysis, generation of *failure reasoning*, and ranking the methods based on their likelihood of causing errors. We will detail this technique below.

### 3.2.1 Generating failure reasoning through call-graph-aware retrieval-augmented (Graph-RAG) code navigation.

The agent first uses code-coverage $C$ to construct the inter-procedural call-graph database $G = (V, E)$, where $V$ is a set of methods in the call-graph, and $E$ is a set of edges representing the caller-callee relationship. This graph is used by the tool-chains, `get_MethodBody` and `get_CallGraph`, to facilitate agent-driven navigation within the call graph. Below is a simplified version of the actual prompt *LLM4FL* uses to drive the agent:

"Given (i) failure reasoning, (ii) stack trace, and (iii) test code, analyze, navigate, and enhance the failure reasoning for the given methods to identify faults. Extract the call graph to get the caller or callee's implementation if more details are needed. Your output should contain: (i) analyzed methods, (ii) enhanced failure reasoning, and (iii) fault ranking."
## **Input**: '{Test Code, Stack Trace, Failure Reason, PrioritizedMethods}'
## **Response**: '{Analyzed Method, Enhanced Failure Reason, Fault Ranking}'

The agent begins the analysis of methods selected sequentially from the prioritized $C'$. The agent retrieves its body using `get_MethodBody` by passing a method identifier and analyzes its implementation for each method. It then queries `get_CallGraph` by passing a method identifier to identify relevant callers or callees, deciding whether to retrieve their implementations to enhance its understanding of code behavior. Throughout the iterative retrieval and analysis process, every method examined, regardless of whether it was prioritized or invoked by a prioritized method, is assigned a failure reason by the agent. This information is saved in a set called $R$. This failure reasoning is a natural language explanation of each method's failure, analyzed in the code navigation process. These results are stored as pairs in a set: $R = \{(m_i, r_i) \mid m_i \text{ is analyzed}, i = 1, \ldots, n\}$. Here, $n$ denotes the number of methods for which failure reasoning was generated.

**Ranking methods based on failure reasoning results.** Finally, given the analysis and failure reasoning $R$, the agent assigns each method $m_i \in \mathcal{R}$ an ordinal rank $(r_i^*)$. Formally, the final ranked set $R^*$ is sorted in descending order based on the ordinal ranks: $R^* = \text{sort}(\{(m_i, r_i, r_i^*) \mid m_i \in R\})$ where $r*_i$ denotes the ordinal rank that reflects each method's likelihood of containing the fault, and $r_i$ explains the reasoning behind the method being faulty. The final ranked list $R^*$ is output in JSON format for subsequent analysis.

## 3.3 Reviewer Agent

Code review is important in software development to ensure software quality. Beyond traditional debugging, code review can help refine failure reasoning and improve fault localization. Inspired by this, we propose *Reviewer Agent*, which uses a novel prompting technique called **Re-ranking through Verbal Reinforcement Learning** to emulate a rigorous code-review process to refine fault localization. We describe the technique below.

### 3.3.1 Re-ranking through Verbal Reinforcement Learning

To refine the fault rankings $R^*$, *Reviewer Agent* adopts a verbal reinforcement learning process inspired by the Reflexion framework (Shinn et al., 2024). The agent iteratively critiques its rank trajectory by identifying inconsistencies, retrieving missing execution insights, and updating method prioritization to better reflect the failure's root cause. After the iterations, a chain of thought is applied to update the final ranking and return the fault localization results. Below is a simplified version of the original *Reviewer Agent*'s prompt:

"Given (i) failure reasoning, (ii) stack trace, and (iii) test code, analyze and critique the initial ranking of suspicious methods. Identify missing or extra details, and refine the analysis using the call graph and method bodies if needed. You should output (1) a revised set of analyzed methods, (2) improved failure reasoning, and (3) final fault ranking with a possible fix."
## **Evaluating the Initial Ranking**: '{Test Code, Stack Trace, Ranked List with Reasonings}'
## **Self-Critique and Refinement**:
- Find missing insights and remove unnecessary information.
- Use 'get_MethodBody' and 'get_CallGraph' to improve reasoning if needed
## **Final Output**: '{Analyzed Methods, Enhanced Failure Reasoning, Fault Ranking}'

Table 2: An overview of our studied projects from Defects4J v2.0.0. *#Faults*, *LOC*, and *#Tests* show the number of faults, lines of code, and tests in each system. *Fault-triggering Tests* shows the number of failing tests that trigger the fault.

| Project | #Faults | LOC | #Tests | Fault-triggering Tests |
|---|---|---|---|---|
| Cli | 39 | 4K | 94 | 66 |
| Closure | 174 | 90K | 7,911 | 545 |
| Codec | 18 | 7K | 206 | 43 |
| Collections | 4 | 65K | 1,286 | 4 |
| Compress | 47 | 9K | 73 | 72 |
| Csv | 16 | 2K | 54 | 24 |
| Gson | 18 | 14K | 720 | 34 |
| JacksonCore | 26 | 22K | 206 | 53 |
| JacksonXml | 6 | 9K | 138 | 12 |
| Jsoup | 93 | 8K | 139 | 144 |
| Lang | 64 | 22K | 2,291 | 121 |
| Math | 106 | 85K | 4,378 | 176 |
| Mockito | 38 | 11K | 1,379 | 118 |
| Time | 26 | 28K | 4,041 | 74 |
| **Total** | 675 | 380K | 24,302 | 1,486 |

**Evaluating and self-critiquing the initial ranking.** The Reviewer Agent uses the initial ranking, $R^*$, provided by the Debugger Agent as the baseline trajectory for refinement. The agent enters a self-reflection phase, guided by Reflexion's Self-Reflection Model ($M_{sr}$), to refine its ranking decisions through verbal reinforcement learning (Shinn et al., 2024). In this phase, the agent examines whether the initial ranking aligns with observed fault rankings by comparing the failure reasoning $r$, with caller-callee interactions and test coverage data, thereby identifying discrepancies such as misordered rankings or missing call dependencies. The agent generates natural language critiques and retrieves additional execution details if needed using tools such as `get_MethodBody` (to obtain complete method implementations) and `get_CallGraph` (to retrieve caller-callee relationships). The refined context is then incorporated in the Reviewer Agent's iterative adjustment process.

**Ranking adjustment through trajectory optimization.** After the initial self-critique phase, the Reviewer Agent refines its initial ranking by updating each method's rank $r^*$ and revising its failure reasoning $r$. Conceptualized as a trajectory optimization problem within the Reflexion framework, the agent, acting as an Actor ($M_a$), leverages reinforcement cues from its Self-Reflection Model ($M_{sr}$) to systematically re-order methods based on the updated evidence of inter-method interactions and failure reasoning. Each iteration integrates new feedback to adjust the ranking until it stabilizes or a preset iteration limit is reached.

**Finalizing the ranking through chain-of-thought.** After the iterative ranking adjustment, the Reviewer Agent finalizes the ranking one last time using chain-of-thought (Wang et al., 2024). To facilitate the thinking process, we ask the Reviewer Agent to generate a probable fix for every method in the final ranking by considering the updated ranking score $R^*$ and refined failure reasoning $R$. After generating all probable fixes, the agent then revisits all the information to do a final re-ranking. Formally, the final ranked list $R^*_{final} = \text{sort}\left(\{(m_i, r_i^*, r_i, f_i) \mid m_i \in C'\}\right)$, where $f_i$ denotes the fix generated for method $m_i$. The final ranked list $R$, encoded in JSON format, provides a structured, machine-readable prioritization along with human-interpretable failure justifications.

## 4 STUDY DESIGN AND RESULTS

In this section, we first describe the study design and setup. Then, we present the answers to the research questions.

**Benchmark Dataset.** We experimented on 675 faults across 14 projects from the Defects4J benchmark (V2.0.0) (Just et al., 2014). Defects4J provides a controlled environment to reproduce faults collected from

various types and sizes of projects. It has been a standard benchmark in automated fault localization research (Lou et al., 2021; Sohn & Yoo, 2017; Chen et al., 2022; Zhang et al., 2017), and more recently, it has been adopted by LLM-based fault localization methods for evaluation (Kang et al., 2024; Qin et al., 2024). Furthermore, recent analysis confirms that Defects4J poses minimal data leakage risk for LLMs (**?**), making it a reliable choice for assessing LLM-based approaches. In our study, we excluded three projects, JacksonDatabind, JxPath, and Chart, from Defects4J because we encountered many execution errors and could not collect test coverage information for them. Table 2 gives detailed information on the projects and faults we use in our study. The faults have over 1.4K fault-triggering tests (i.e., failing tests that cover the fault). The sizes of the studied projects range from 2K to 90K lines of code. Note that since a fault may have multiple fault-triggering tests, there are more fault-triggering tests than faults.

**Evaluation Metrics.** According to prior findings, debugging faults at the class level lacks precision for effective location (Kochhar et al., 2016). Alternatively, pinpointing them at the statement level might be overly detailed, omitting important context (Parnin & Orso, 2011). Hence, we perform our fault localization process at the method level in keeping with prior work (Benton et al., 2020; B. Le et al., 2016; Li et al., 2019; Lou et al., 2021; Vancsics et al., 2021). We apply the following commonly used metrics for evaluation:

*Recall at Top-N*. The Top-N metric measures the number of faults with at least one faulty program element (in this paper, methods) ranked in the top N. The result from *LLM4FL* is a ranked list based on the suspiciousness score. Prior research (Parnin & Orso, 2011) indicates that developers typically only scrutinize a limited number of top-ranked faulty elements. Therefore, our study focuses on Top-N, where N is set to 1, 3, 5, and 10.

**Implementation and Environment.** To collect test coverage data and compute results for baseline techniques, we utilized Gzoltar (Campos et al., 2012), an automated tool that executes tests and gathers coverage information. For the LLM-based components, we employed OpenAI's gpt-4o-mini-2024-07-18, a more cost-effective yet capable LLM (OpenAI, 2024). We used LangChain to develop *LLM4FL* (Langchain, 2024). We designed the prompts to be concise to minimize token usage and to allow more room for analysis-related information and code. Our code and prompts are available online (AnonymousSubmission, 2025). To reduce the variations in the output, we set the temperature parameter to 0 during model inference.

**RQ1: How does *LLM4FL* perform compared with other fault localization techniques?**

**Motivation and Approach.** We compare *LLM4FL*'s fault localization accuracy with five baselines representing different methodological families: ***Ochiai*** (statistical) (Abreu et al., 2006), ***DeepFL*** (deep neural network) (Li et al., 2019), ***Grace*** (graph neural network) (Lou et al., 2021), ***DepGraph*** *(graph neural network)* (Rafi et al., 2024), ***AutoFL*** *(LLM)* (Kang et al., 2024), and ***AgentFL/SoapFL*** (Qin et al., 2024) (LLM).

*Ochiai* (Abreu et al., 2006) is a widely recognized statistical fault localization technique known for its high efficiency, making it a common baseline for comparison (Lou et al., 2021; Li et al., 2021; Cui et al., 2020; Wen et al., 2019; Qian et al., 2021; Rafi et al., 2024). As such, we use *Ochiai* to rank the methods during the segmentation process and include it as a baseline for accuracy comparison.

*DeepFL* (Li et al., 2019) is a deep-learning-based fault localization technique that integrates spectrum-based and other metrics such as code complexity, and textual similarity features to locate faults. It utilizes a Multi-layer Perceptron (MLP) model to analyze these varied feature dimensions. We follow the study (Li et al., 2019) to implement *DeepFL* and include the SBFL scores from 34 techniques, code complexity, and textual similarities as part of the features for the deep learning model. *Grace (Lou et al., 2021)* utilized graph neural networks (GNN). It represents code as a graph and uses a gated graph neural network to rank the faulty methods. *DepGraph* (Rafi et al., 2024) is another GNN-based technique that further improves *Grace* by enhancing code representation in a graph using interprocedural call graph analysis for graph pruning and integrating historical code change information.

*AutoFL* (Kang et al., 2024) is an LLM-based fault localization approach that provides the LLM with a failing test and method descriptions to gather relevant coverage data, then repeats the process to assign inverse scores and rank candidate methods by averaging results across runs. *SoapFL* (also known as *AgentFL*) (Qin

Table 3: Fault localization accuracy (Top-1, 3, 5, and 10) for 675 faults from Defect4J V2.0.0. The numbers in the parentheses show the percentage difference compared to LLM4FL.

| Techniques | Top-1 | Top-3 | Top-5 | Top-10 |
|---|---|---|---|---|
| Ochiai | 121 (169.42%) | 260 (63.46%) | 340 (39.41%) | 413 (20.10%) |
| DeepFL | 257 (26.85%) | 353 (20.40%) | 427 (11.01%) | 468 (5.98%) |
| Grace | 298 (9.40%) | 416 (2.16%) | 486 (-2.47%) | 541 (-8.32%) |
| DepGraph | 359 (-9.19%) | 481 (-11.64%) | 541 (-12.38%) | 597 (-16.92%) |
| AutoFL | 275 (18.55%) | 393 (8.14%) | 423 (12.06%) | 457 (8.53%) |
| SoapFL/AgentFL | 311 (4.82%) | 414 (2.66%) | 455 (4.18%) | 478 (3.77%) |
| LLM4FL | 326 | 425 | 474 | 496 |

et al., 2024) organizes fault localization into structured phases, including test failure comprehension, code navigation, and iterative LLM-based scoring, leveraging enhanced method documentation to refine suspicious methods. While both *AutoFL* and *SoapFL* used OpenAI's GPT-3.5 for their experiments, for our evaluation, we adapted both techniques to use the same LLM version as *LLM4FL* (i.e., gpt-4o-mini-2024-07-18) for comparison.

**Results.** ***LLM4FL outperforms the LLM-based baselines, AutoFL and AgentFL, by achieving a 185.55% and 4.82% improvement in Top-1, respectively.*** Tables 3 show the fault localization results of *LLM4FL* and the baseline techniques. Among the three LLM-based techniques, *LLM4FL* achieves a better Top@N across all values of N. In the Top-1 metric, *LLM4FL* locates the correct fault in 326 cases, compared to *AutoFL*'s 275 and *AgentFL*'s 311, representing an 18.55% and 4.82% improvement, respectively. Similarly, in Top-3, Top-5, and Top-10, *LLM4FL* an improvement between 8.14% to 12.06% over AutoFL and 2.66% to 4.18% for *AgentFL*. These numbers highlight *LLM4FL*'s ability to pinpoint faulty methods more accurately. For cost, we compute the total dollars spent by multiplying each approach's token usage by the per-million-token price (e.g., $0.150 for input, $0.600 for output). *SOAPFL averages about $0.055 per bug, AutoFL averages about $0.065, while LLM4FL is at around $0.050 per bug making our approach comparably cost-effective.*

***LLM4FL shows higher Top-1 and Top-3 compared to most other non-LLM-based techniques.*** For the Top-1 metric, *LLM4FL* scores 326, which is 169.42% higher than Ochia's score of 121, 26.84% higher than DeepFL's score of 257, and 9.39% better than Grace's result of 298. One exception is *DepGraph*, which achieves a Top-1 of 359, 8.64% higher than *LLM4FL*. As the range expands to Top-3 and beyond, *LLM4FL* demonstrates its robustness, significantly outperforming DeepFL and maintaining strong performance alongside Grace. LLM-based techniques have several advantages over traditional techniques such as *DepGraph*. First, techniques like *LLM4FL* leverage pre-trained LLMs in a zero-shot setting, which can be easily applied to systems with insufficient training data. Second, techniques like *DepGraph* require additional model training that can take days. LLM-based techniques leverage generic pre-trained LLMs without specific fine-tuning or re-training. Finally, LLM-based techniques can explain the decision, which helps with adaption (Kang et al., 2024; Qin et al., 2024). Hence, LLM-based techniques have strong potentials to enhance fault localization by offering greater adaptability, reducing training overhead, and providing interpretable explanations.

*LLM4FL* achieves a higher Top-1 compared to other LLM-based techniques, *AutoFL* and *AgentFL*, by 18.55% and 4.82%, respectively. It also has competitive results compared to supervised techniques like *DeepFL* and *Grace*, leveraging a pre-trained LLM in a zero-shot setting.

**RQ2: Does order matter in the initial list of methods provided to the LLM?**

**Motivation.** *LLM4FL* divides the coverage data into different divisions using an order-aware division strategy to address the token size limitation of LLMs. We sort the methods using the *Ochiai* scores before the division, though different sorting mechanisms may affect the final fault localization result. Although *LLM4FL* eventually visits and assesses every method, a recent study (Chen et al., 2024a) observes that the

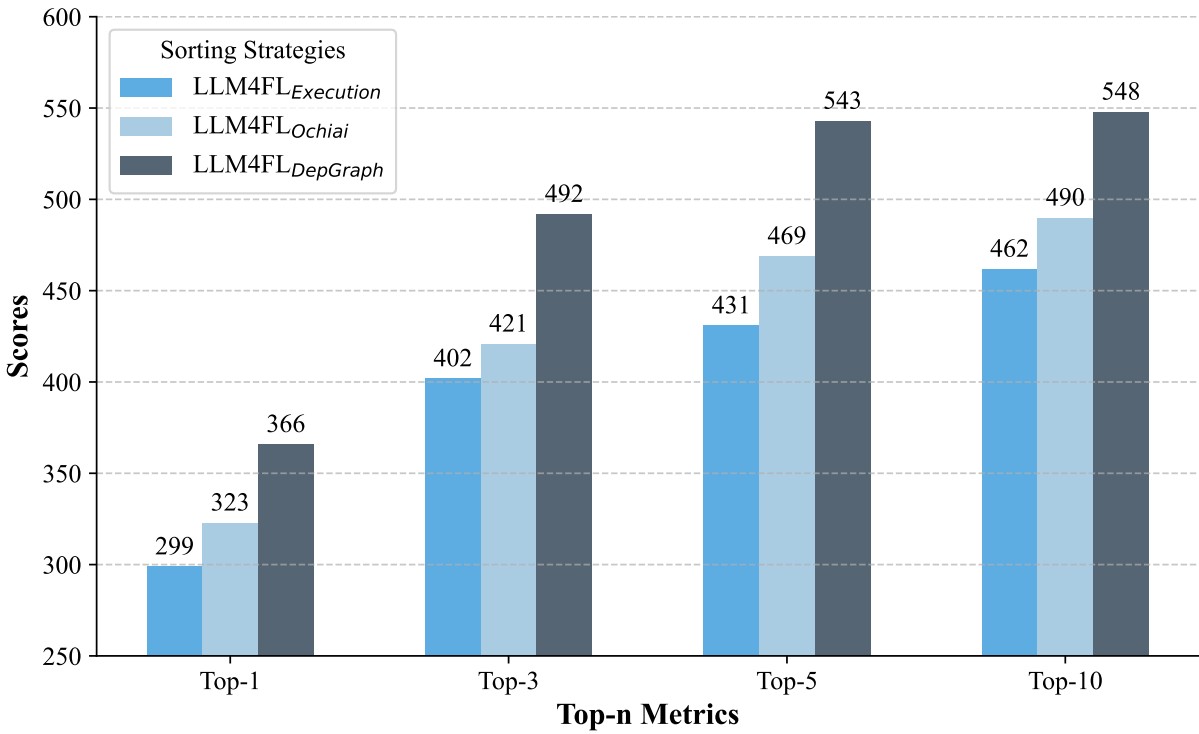

Figure 3: Fault localization results when using different method sorting strategies during the order-aware division process.

order of premises affects LLM's results. However, whether this effect extends to software engineering tasks, particularly fault localization, remains unclear. Hence, in this research question, we investigate whether the order of methods within the groups affects the LLM's fault localization performance.

**Approach.** To test the effect of method ordering, we experiment with three distinct sorting strategies: $LLM4FL_{Execution}$, $LLM4FL_{Ochiai}$ (the default sorting in $LLM4FL$), and $LLM4FL_{DepGraph}$ to sort the methods before the divide-and-conquer step.

$LLM4FL_{Execution}$: We use the unsorted list of methods executed during testing, as generated by Gzoltar (Campos et al., 2012). This default list represents the natural execution order of the methods, with no explicit ranking or prioritization. By providing the LLM with methods based on the execution sequence, we establish a control case to measure its performance without any ranking influence.

$LLM4FL_{Ochiai}$: As discussed in Section 3.1, we apply *Ochiai* to sort the methods before the divide-and-conquer process. *Ochiai* is unsupervised and is efficient to compute. We hypothesize that providing the LLM with methods sorted by their suspiciousness score will lead to more effective fault localization, as the model will focus on the most likely faulty candidates earlier in the process.

$LLM4FL_{DepGraph}$: It uses the ranking produced by *DepGraph*, a state-of-the-art Graph Neural Network (GNN)-based fault localization technique (Li et al., 2015; Rafi et al., 2024), to sort the methods. *DepGraph* ranks methods based on structural code dependencies and code change history. As shown in RQ1, *DepGraph* shows the highest fault localization accuracy among all the techniques, surpassing $LLM4FL_{Ochiai}$. By examining the fault localization result after sorting the methods using *DepGraph*'s scores, we can better study if the initial order affects LLM's results, even though LLM eventually visits all the methods.

**Results.** *Method ordering has a significant impact on LLM's fault localization result, with up to 22% difference in Top-1 (from 299 to 366).* Figure 3 shows the fault localization results using different

sorting strategies. When methods were presented in the execution order, $LLM4FL_{Execution}$ achieved a Top-1 score of 299, 402 for Top-3, 431 for Top-5, and 462 for Top-10. This performance establishes a baseline, showing how the LLM behaves without strategic ordering. However, sorting methods with the lightweight *Ochiai* scores resulted in noticeable improvements across all Top-N, where $LLM4FL_{Ochiai}$ improved the Top-1 score to 323, an 8% increase over $LLM4FL_{Execution}$.

***$LLM4FL_{DepGraph}$ provides further improvement to the already-promising result of DepGraph, indicating method ordering is critical to LLM4FL, or LLM-based fault localization in general.*** $LLM4FL_{DepGraph}$ achieved the highest Top-1 score of 366, which significantly outperforms both LLM4FL_{Execution} and LLM4FL_{Ochiai}. The improvement was consistent across all the metrics. We also see that $LLM4FL_{DepGraph}$ has better Top-1, 3, and 5 scores compared to *DepGraph*. This consistent improvement underscores the importance of method ordering in enhancing the accuracy of LLM-based fault localization. Namely, if the initial order is closer to the group truth, the final localization result tends to be more accurate.

Our finding establishes a new research direction for LLM-based fault localization, or any software engineering tasks that take a list of software artifacts as input. Future studies may study how different premises of ordering affect other software engineering tasks, and how to combine traditional software engineering techniques to pre-process LLM's input to improve the results further.

> The initial method ordering significantly impacts the accuracy of LLM-based fault localization, with Top-1 scores varying by up to 22%. **Combining *LLM4FL* and DepGraph further improves the results**. Future research should explore various ordering strategies and how traditional software engineering techniques can be integrated to optimize LLM performance further.

### RQ3: How do different components in *LLM4FL* affect the fault localization accuracy?

**Motivation.** *LLM4FL* employs several components, each of which plays a distinct role in the overall process. Hence, in this RQ, we conduct an ablation study by removing each component separately and studying their impact on fault localization accuracy. The findings may inspire future studies on adapting the components for similar tasks.

**Approach.** To evaluate the impact of each component, we designed four different configurations:

$LLM4FL_{w/o\ CodeNav}$ removes the code navigation mechanism, meaning the Debugger agent no longer does fault navigation by retrieving the call graphs. Instead, the *LLM4FL* uses a single prompt to perform fault localization without fault navigation. This configuration tests whether using the caller-callee information improves the ranking and selection of faulty methods.

$LLM4FL_{w/o\ Division}$ removes the order-aware division of the covered methods, giving the agents the entire coverage data at once instead of dividing it into smaller, manageable groups. Coverage segmentation addresses token limitations in LLMs, so removing it explores the impact of feeding the full dataset to the agents in one step. We aim to see how handling large amounts of data in a single input influences the fault localization result, as it may overwhelm the model or reduce precision.

$LLM4FL_{w/o\ Reflexion}$ removes the verbal reinforcement learning technique, which is used to allow agents to review and refine their initial ranking. Without this step, the agents rely solely on their initial assessments without iterative improvements.

**Results.** *While all components help improve the results, including coverage division and code navigation, provide the largest improvement to fault localization results (23% and 17% in Top-1).* Table 4 shows the Top-1, 3, 5, and 10 scores when each component is removed. Removing coverage divisions has the largest overall impact across all scores, reducing Top-N by 19% to 23%. Removing prompt chaining has the second largest impact (11% to 17%). At the individual project level, these two components also have the largest impact in Top-1 in 9/13 studied projects. Our finding shows that employing sorted coverage grouping following the divide and conquer technique and agent communication significantly

Table 4: Impacts of removing different components in *LLM4FL* on Top-1, 3, 5, and 10. The numbers in the parentheses show the percentage changes compared to *LLM4FL* with all the components.

| Techniques | Top-1 | Top-3 | Top-5 | Top-10 |
|---|---|---|---|---|
| LLM4FL | 326 | 425 | 474 | 496 |
| LLM4FL$_{w/o\ CodeNav}$ | 273 (-16.51%) | 378 (-11.06%) | 409 (-13.53%) | 409 (-17.21%) |
| LLM4FL$_{w/o\ Division}$ | 251 (-23.24%) | 341 (-19.76%) | 365 (-22.83%) | 381 (-22.87%) |
| LLM4FL$_{w/o\ Reflexion}$ | 290 (-11.31%) | 400 (-5.88%) | 436 (-7.82%) | 459 (-7.09%) |

improves fault localization results. Future research should consider these techniques when designing fault localization techniques.

***Although there is no oracle during the fault localization process, asking LLMs to self-reflect still helps improve the overall Top-1 by 11%.*** LLMs often suffer from hallucinations, especially when there is a lack of feedback from external oracles (Xu et al., 2024; Huang et al., 2023). Even though we did not provide any ground truth or external feedback to LLM, we found that Reflexion is still effective in improving fault localization results. We speculate that verbal reinforcement learning helps the model improve its results by revisiting the suspicious methods it ranked earlier, creating a feedback loop. In this process, we analyze the results for each group of methods and combine these group-specific results into a final ranking. The model can then spot mistakes or gaps in its logic, leading to better results. Self-reflection brings 6% to 11% improvement across the Top-N metrics. Our finding suggests that future studies should consider self-reflection even without external feedback. Our finding highlights the effectiveness of self-reflection, which should be considered in future fault localization results.

> The results show that each component of *LLM4FL* contributes to its overall fault localization performance, with **coverage division and code navigation having the largest positive impact**. Removing these components leads to significant declines in accuracy, confirming their role in finding faults in a large code base.

## 5 Threats to Validity

**Internal Validity.** A potential threat to internal validity is the risk of data leakage in large language models (LLMs), where the model might have been exposed to the benchmark data during training. Nevertheless, Ramos et al. (Ramos et al., 2024) found that newer and larger models trained on larger datasets exhibit limited evidence of leakage for defect benchmarks. Since we utilize GPT-4o-mini, a large model trained using a tremendous amount of data, it likely shares similar characteristics in reducing memorization risks. Additionally, following a prior work we ensure no content related to the project name, human-written bug report, or bug ID is entered into ChatGPT to minimize the risk of data memorization (Qin et al., 2024).

**External Validity.** Our evaluation is based on Defects4J, a well-established dataset in the software engineering community. Although this dataset includes real-world bugs, the systems studied are primarily Java-based. Future studies may extend our study to other programming languages or domains.

**Construct Validity.** Construct validity relates to whether the metrics we used accurately measure the performance of fault localization techniques. We used widely accepted Top-N metrics, which are commonly utilized in prior fault localization studies. However, our results are based on the assumption that developers primarily focus on the top-ranked faulty methods. Although this assumption aligns with previous research, different development practices could influence the effectiveness of our approach.

## 6 Conclusion

In this paper, we introduced *LLM4FL*, an LLM-agent-based fault localization approach. It utilizes multiple specialized LLM agents, including Context, Debugger, and Reviewer, to iteratively refine and improve the accuracy of fault localization through order-aware prioritization, graph-based code navigation, and self-

reflection using verbal reinforcement learning. Evaluated on the Defects4J (V2.0.0) benchmark, *LLM4FL* demonstrated significant improvements over existing approaches, achieving an 18.55% increase in Top-1 accuracy compared to AutoFL and 4.82% improvement over SoapFL. Further enhancements, including coverage segmentation and iterative refinement, increased accuracy by up to 22%. Future work will explore expanding *LLM4FL*'s capabilities for larger and more diverse codebases, further refining the agent collaboration and reasoning mechanisms.

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
