# OpenReview forum: "LLM4FL: Multi-Agent Repository-Level Software Fault Localization via Graph-Based Retrieval and Iterative Refinement"
_TMLR — Rejected by TMLR_

### Review · Reviewer_uTz5 · 2025-09-13

**Summary Of Contributions:**

Summary:

The work proposes an LLM-driven system for localizing faults in a code base where faults are test
failures and localization means finding the method(s) responsible. The inputs the system uses
include the code making up the code-base and fault, as well as the call graphs/traces and code
coverage information (which code is executed or not during the tests).

This is processed via three LLM-based components (or "agents") that interact to solve the problem.
The first agent is tasked with describing the failure from the test source and trace. It also
produces a list of methods to inspect further ordered by way of an existing coverage-based
technique (SBFL, unsure whether an LLM is involved in producing the ordering but ordering is
important for inputs to the LLMs). Part of the motivation for this agent is limits on LLM context
size so the agent does a lot of pruning of irrelevant information (e.g. calls to external
libraries) and dividing its task (or context) across multiple LLM calls. The second agent navigates
the call graph to rank candidate methods as prioritized by the first agent. It is also allowed to
make adjustments to the failure description and produces a ranking of methods deemed responsible.
The final agent does some reasoning with itself based on the second's result to adjust the ranking.
The end result is a ranking of methods, hopefully with the top method(s) being responsible for the
test failure.

Experiments demonstrate the system on an Java faults dataset from 2014 with the new system (LLM4FL)
running using gpt-4o-mini. Several baseline methods are tested alongside, with examples making use
of the same raw data that LLM4FL uses such as call graphs. The results show LLM4FL is better at
finding the faults than 5 out of the 6 baseline methods. The one method (DepGraph) that beats the
proposed method seems to also incorporate historical code change information which LLM4FL does not
use, and requires training a (graph) neural network for the task.

Other than the system overall accuracy, some experiments demonstrate the importance and value of
order in processing methods to rank and the relative importance of the three agents making up the
proposed system.

Strengths:

+ S1. Comparison against multiple baselines across several categories. While not the producing the
  best results among baselines, it is better than methods that do not require training models for
  the task.

Weaknesses:

- W1. Lack of scientific content. The primary issue with this work and why I do not think the
  current version belongs in a scientific journal is that it lacks scientific content. By this I
  mean conclusions that both 1) are sufficiently supported by sound arguments and/or experiments,
  and 2) are at least minimally generalizable to some topic.

  The current article fails in these regards primarily due to the many confounding factors that
  prevent its conclusions from being generalizable, or correct as stated given statements are quite
  general. The system employs a particular LLM for most of its operation with prompts describing
  the various tasks and an LLM carrying it out with optional aid of code navigation tools and other
  sources of data. LLM operation is non-deterministic even under intentionally limited uses and
  LLMs differ vastly in capability and quirks. Further non-determinism due to agents architecture
  makes it difficult to trust any number or conclusion is representative or generalizable to
  outside a very narrow scope. The scope for which current results can be applied to is so narrow
  that it is difficult to imagine who would be able "interested in knowing the findings of this
  paper". This applies to the RQs and other points throughout the article.

  I elaborate on the lack of support for claims in the next section and offer some options on
  revising the work along two avenues in the "Requested Changes" section which I would read as
  "Suggested Changes".

- W2. Significant lack of contribution-relevant details. There are major design choices or aspects
  of the proposed system which lack detail and/or motivating experimentation:

  - Cost and/or time comparison with baselines. The baselines that are primarily based on calls to
    LLMs are described using the API usage costs but one cannot compare the non-API computational
    requirements or the overall requirements (cost or time) of LLM4FL relative to all other
    baselines or even the top performing baseline. DepGraph presumably requires a different type of
    effort or cost than calls to an LLM service but how do those costs compare? Can LLM costs and
    other costs be put on the same scale by considering the computational cost of a locally
    deployed LLM of a similar performance to the OpenAI one? Are there reasons why an analyst
    looking to deploy a fault localization system would not use the top performer, DepGraph?

  - Agent interactions. The system organization as shown in Figure 1 seems mostly unidirectional
    handoff of results from the first to last agent. I cannot tell whether there is any part of the
    system that does not follow a sequential process of "Context Extractor Agent" does a bit of
    work and hands off full results to "Debugger Agent" which then does a bit of work and hands off
    the complete result to "Reviewer Agent". If there are other interactions, no details of such
    interactions are presented and if there are no non-sequential interactions, why is the
    agent-based organization used to begin with?

    Plausibly, each phase may be sequential, but may be allocated different amount of reasoning
    time in which case this allocation needs to be explored experimentally.

  - Context splitting and trace pruning. The description of the large context splitting as given in
    3.1.1 just states that methods are not split across requests but says nothing about the
    divide-and-conquer aspect of the approach other than the use of an ordering system based on
    coverage-test statistics. If there is nothing in the methodology beyond not splitting methods,
    I don't think it would constitute a significant enough choice to mention beyond a passing note.
    If it is a significant choice, experiments comparing against the opposite would be warranted.
    For the trace-pruning, little is said about how irrelevant stack frames are determined to be
    irrelevant (other than those into external libraries). Does this only apply to external
    libraries?

  The comments above hint as to how to address these issues and I offer further suggestions in the
  sections below.

**Additional Comments:**

- C1. I would find it surprising that modern LLMs did not include training data from a code dataset
  from 2014. The fact that the reference claiming this is not the case is missing in the paper is
  troublesome.

- C2. The example fault description in Section 3.1. seems odd in that the fault is in the test and
  not the code (if I read that right). If this is a sufficiently common situation, it might be
  useful to separate out such faults in the results. I cannot tell whether this makes them easier
  or harder to localize though. For a human, I'd say easier, but unsure about the LLM-based scheme.

Questions:

- Q1. Experiments excluded several projects from the dataset because "many execution errors". How
  different are execution errors from faults, which I understand is the main subject of the
  dataset?


Smallest comments:

- Missing reference "A recent study (?) also shows that it has a very low risk of data leakage
  ...". The same reference shows up later in the paper as (?) again.

- Typo at "185.55%", the number in the table is "18.55%".

- Typo at "8.64%", the number in the table is "-9.19%".

- Typo at "group truth".

**Audience:**

No

**Audience Explanation:**

See weakness W1.

**Broader Impact Concerns:**

Not needed.

**Claims And Evidence:**

No

**Claims Explanation:**

Most of the lack of support is a consequence of weakness W1. There is far too much non-determinism
and other confounders to make conclusions. All research questions are affected by this. I cannot
tell whether the reported results will remain consistent were the experiment done again or using a
different LLM. I elaborate more on the RQs in the next Requested Changes section below.

**Requested Changes:**

I organize my recommendations into two categories: the more expansive are changes needed for
publishing the work as a scientific paper and the lesser are changes needed for publishing the work
at a more specialized venue or track. Both options need to address W2 but W1 is critical for a
scientific journal.

- A) To publish this work in a primarily scientific venue I suggest that both W1 and W2 need to be
  addressed. Specific to W1 I recommend, I first recommend controls for non-determinism and the LLM
  choice:

  - Systematically account for and control for non-determinism. Statistical validity analysis can
    help tackle this issue. This can be done by running non-deterministic components repeatedly and
    reporting the mean measurements alongside measures of dispersion (IQR, 95% confidence margins,
    etc.).

  - Account for and control for LLM, its configuration, and "seeming trivia" (to be explained
    below). Currently a single LLM is tested with. How robust is the system to the LLM choice? Are
    there minimal requirements that an LLM must meet in order to be useful as part of LLM4FL
    (perhaps tuned for thinking segments?) How robust is the system to choices of things that seem
    trivial like prompts how they are written? The same instruction can be given in numerous ways
    but presumably only one is tested with. Do I need to copy it exactly for it to work as well as
    presented in my deployment?

  Beyond this, I offer recommendations across various experiments largely by way of questions that
  need to be answered in order to support conclusions made. When writing a conclusion about
  fault-localization or even LLM-based fault-localization, some care needs to be made as to
  properly accounting for the generality of the statement as otherwise the conclusions need to be
  restated as about LLM4FL in particular.

  - Comparisons (RQ1): Here accounting for non-determinism and foundation model selection (for the
    baselines that apply) would go a long way. Additionally, some care needs to be taken as to what
    degree each baseline is allowed to be optimized in terms of its configuration/parameter choices
    for the task. Related, can other LLM-based techniques make use of the same ideas that were
    presented in LLM4FL (and vice versa)?

  - Sorting Strategies (RQ2): The interaction between sorting strategy and LLM4FL is presented but
    is the observation expected were the different strategies used alongside other systems? That
    is, are the differences in Figure 3 due to the sorting strategies, due to LLM4FL, or to what
    extent is it the effect due to each?

  - Ablations (RQ3): It is good that ablation experiments are performed but they granularity of the
    experiments is too coarse. There are doubtless many parameters or configurations underlying
    each of the three ablated components but only their use as a whole was tested. Can they be
    misconfigured to produce results worse if they are used? Furthermore, the interactions between
    the components is not experimented with. Do the losses in Top-1 accuracy stack linearly were
    multiple components ablated?

  I believe there is plenty of exploration to be made around the system and were the reasoning and
  conclusions made with more care, the paper could live up to its goal I find nicely worded on page
  14:

   "Our finding establishes a new research direction for LLM-based fault localization, or any
   software engineering tasks that take a list of software artifacts as input."

  Suggestions above would bring the work closer to the first and I hope the authors further improve
  the work to be able to confidently state the second.

- B) The work may be better served by a narrower or less scientific-focused venue like an industry
  track at AI conference or a very specific workshop. In these settings, the generalizability of
  conclusions need not be wide and often a demonstration is sufficient for publication. Despite
  this, weakness W2 still need to be addressed. To that end, I recommend elaborating on the details
  of the design choices I outlined above and if possible, answer the questions I posed there. W1 is
  not critical for such a venue though it would be great to see proper controls for non-determinism
  in experimental results so I recommend that regardless.

---

> ### Author Response · Authors · 2025-09-30
> **Robustness, Cost/Time Trade-offs, Agent Design, Context Splitting, Stack-Traces, and Execution Errors**
>
> Thank you for your detailed feedback. Below, we provide our responses to the main points raised.
> # Non-determinism & robustness
> We thank the reviewer for raising this important point. To account for the stochastic nature of LLMs, we reran all LLM-based techniques with three different random seeds and report the averages along with standard deviations. The results show very limited variation across runs (≈1%), indicating that our conclusions are stable. For example, LLM4FL achieves a Top-1 accuracy of 326.3 ± 3.8 and a Top-3 of 424.7 ± 3.2, which are consistent with the originally reported numbers. We will update Table 3 and the text to clarify that results are averaged across multiple runs and to explicitly include the variance.
>
> # LLM4FL vs. DepGraph
> DepGraph outperforms LLM4FL on some Top-N metrics, but it is a supervised method that requires project-specific training data and a GNN architecture, both of which add significant cost and complexity. LLM4FL, in contrast, is a zero-shot approach requiring no task-specific training, making it more easily deployable. More importantly, when we combine LLM4FL with DepGraph ordering, the combined approach surpasses when using DepGraph alone, highlighting that LLM4FL complements, rather than replaces, non-LLM and supervised ML techniques.
>
> # Cost Analysis
> We thank the reviewer for this comment. We agree that costs and time are important, but also note that direct comparisons across all techniques are not entirely possible, as the actual runtime depends heavily on hardware, deployment setup, and the availability of project-specific training data. For example, DepGraph and Grace require project-specific bugs and labeled history for training, which some systems may not have. Moreover, their training costs (GPU hours, wall-clock time, peak memory) vary by hardware. To provide more context, on the Lang project (64 bugs) (a 40G NVIDIA A100 GPU), Grace training took ~30 minutes and DepGraph ~12 minutes (with inference time of around ~52s and ~29s per bug, respectively). In contrast, LLM4FL is zero-shot and requires no training data, and takes around ~20s per bug on a regular laptop.
>
> More importantly, when combined with DepGraph ordering, LLM4FL surpasses DepGraph alone, showing complementarity. In the revision, we will also report running times for LLM-based and, where possible, traditional techniques.
>
> # Agent interactions (is it just a sequential handoff?)
> Our pipeline has three stages (Context -> Debugger -> Reviewer), but each stage loops internally rather than doing a one-shot handoff. For example, the Debugger can pull additional code by following caller/callee links when it needs more code context, and the Reviewer can revisit or call additional method bodies or call-graph details before re-ranking. We structured it this way to separate concerns. The Context agent trims noisy inputs under token limits, the Debugger focuses on structural navigation, and the Reviewer provides the final ranking. This design tries to follow a natural debugging flow: triage the failure context, investigate dependencies, and then review and refine the diagnosis.
>
> # Context Splitting Significance
> The context splitting in LLM4FL is not only about keeping methods intact across requests but also about applying an order-aware divide-and-conquer strategy. Methods are first ordered using SBFL scores, then partitioned into token-bounded groups, and each group is analyzed separately before results are merged. This is to look for failure context even when the coverage data is large and the higher-suspicion methods appear earlier in the LLM’s reasoning. The impact of this design is shown in RQ2, where different orderings (Execution, Ochiai, DepGraph) lead to differences of up to 22% in Top-1 accuracy. Furthermore, in our ablation (Table 4), removing division altogether caused a 23% drop in Top-1 accuracy, confirming that the order-aware division is a significant contributor. We will clarify this rationale more explicitly in Section 3.1.1.
>
> # Stack-trace pruning?
> Thanks for your question. We do not just filter “external libs.” We label each frame as in-project or out-of-project using a project-specific allowlist of package prefixes (from the repo/source tree) and file-path mapping. We then anchor at the deepest in-project frame and drop all frames below it, since those are test/runtime machinery (e.g., JUnit, reflection) that explain how the error surfaced, not where the fault likely is. To retain a useful signal, we keep pre-anchor test helpers reachable via the static call graph.
>
> # Execution errors in some projects
> Thanks for your question. We could not generate dynamic coverage due to persistent build failures. Many buggy versions of the project failed to compile despite our best efforts to resolve issues such as incompatible JDKs, outdated configurations, or unavailable external dependencies. We will clarify this in the revised version.

---

> > ### Comment · Reviewer_uTz5 · 2025-10-16
> > **Re: non-determinism**
> >
> > Author comment suggests at least some of the non-determinism in the approach is not impactful. What about non-determinism in LLMs? Has temperature above 0.0 been tested to see if the LLMs are consistently effective?

---

> > ### Comment · Reviewer_uTz5 · 2025-10-16
> > **Re: cost analysis and LLM choice**
> >
> > Author response stated "In contrast, LLM4FL is zero-shot and requires no training data, and takes around ~20s per bug on a regular laptop.". This is assuming the regular laptop is making API calls to an external LLM running on unknown hardware. What if it were using the same 40G NVIDIA A100 GPU? Can you estimate the cost of running the best comparable open model on this or other hardware, and making the necessary queries on that hardware?
> >
> > A related question is to what degree does the approach depend specifically on the chosen closed-source LLM and the related question of how much tuning (prompts, other parameters) must the method go through to apply to a particular LLM?

---

### Review · Reviewer_as3c · 2025-09-15

**Summary Of Contributions:**

The paper proposes LLM4FL, a novel multi-agent system to localise faults in code bases. The proposed framework relies on three LLM-based agents to analyse the code and rank possible faulty methods. The authors test the proposed approach against a few LLM-based and non-LLM-based baselines on the Defects4J dataset. While LLM4FL outperforms other LLM-based approaches, it fails to reach state-of-the-art performance. Moreover, the proposed approach is tested on a single dataset and only on Java code, thus failing to ensure its reliability and generalizability. Additionally, a few elements are missing from the framework description, such as the prompt engineering process. Therefore, I believe that the current manuscript does not meet the quality standards of TMLR.

**Additional Comments:**

- The paper contains undefined references such as “A recent study (?)” in the introduction and “Defects4J poses minimal data leakage risk for LLMs (?)” in section 4. Please fix them.
- While the proposed methodology based on a multi-agent framework is clearly presented and reasonable, I do not understand the reason why the authors consider adding the final ranking through chain of thought process. It seems that the ranking has already been analysed multiple times using the trajectory optimisation process relying on self-reflection. Therefore, it seems that the final chain of thought process is quite superfluous. Could the authors provide any insight into why such a final step is made necessary?
- LLM4FLOchiai is presented as the default sorting in LLM4FL. Meanwhile, LLM4FLDepGraph seems to be the best-performing sorting approach. Therefore, I wonder why the authors did not consider making LLM4FLDepGraph the default sorting mechanism.
- The set of references used in the first paragraph of the introduction is slightly outdated. This might be an indication that the task the authors are trying to solve has not been explored recently. Is this the case? If not, can the authors provide more recent pointers to the state-of-the-art literature in the context of software fault identification?

**Audience:**

Yes

**Audience Explanation:**

I believe that the paper targets an interesting topic of research, and the idea of defining a multi-agent system to tackle fault identification is interesting. Therefore, I do believe that the paper topic might be of interest to some of the TMLR’s readers. My issues concerning the paper do not regard its topic or its connection to the TMLR community.

**Broader Impact Concerns:**

I do not have any concerns about the ethical implications of this paper. As the paper provides an automatic system to identify faults in code bases, I think it can only contribute positively to society.

**Claims And Evidence:**

No

**Claims Explanation:**

LLM4FL outperforms only other LLM-based approaches, while it fails to reach state-of-the-art performances. DepGraph seems to be far more effective in identifying faults while also being more generalizable. Moreover, while the authors mentioned a few times in the paper that LLM4FL is reliable and effective, the proposed approach is tested only on a single dataset and only on Java code. Its reliability and generalizability seem to be assumed without thorough testing.

**Requested Changes:**

- The paper contains undefined references such as “A recent study (?)” in the introduction and “Defects4J poses minimal data leakage risk for LLMs (?)” in section 4. Please fix them.
- What is an SBFL formula? It was used in the introduction to elicit the paper’s contributions without providing its definition.
- The related work section has a related work subsection. This is quite confusing, and I think that it is an oversight made by the authors. I’d suggest fixing this issue.
- I’d suggest the authors add an overview of LLMs applications outside the context of code processing and fault identification, to better introduce why it might be relevant to use LLMs. The same applies to the LLM agents' definition. For example, the authors might consider mentioning LLMs as ontology construction tools [1], LLMs for plan generation [2], and LLMs for financial decision making [3]. Similarly, whenever dealing with LLMs (and ML in general), I think it is worth noting the explainability issue, meaning that it is complex to understand how such LLMs reason and if their predictions will be reliable. In this context, I’d suggest that the authors add a brief discussion to the paper, mentioning, for example, [4,5,6].
- The authors should consider making Table 1 less verbose. The table contains the information required to understand the limitations of the current literature and, as such, is relevant. However, it is too verbose and basically iterates once again what is written in the introduction and the related work section. Therefore, it should be shrunk to give a summary of each limitation, without describing them in detail. The authors might also consider converting the table into a checkmark table, like Table 4 in [7], where each “feature” is either satisfied or not by the approaches available in the state-of-the-art.
- While the proposed methodology based on a multi-agent framework is reasonable, the authors fail to provide an intuition or an explanation on why such a framework is designed in the way it is. Why did the authors consider leveraging three agents? Why are the components of each agent defined in the way they are? These represent very valuable questions that should be addressed in the paper.
- Similarly to the above, the authors fail to provide any insight into how they defined the prompts that were used. Prompt engineering represents a well-established paradigm in LLM research, and it has been proven that different prompts may significantly alter the performance of the LLM on a given task [8]. Therefore, the authors should consider giving more details about this aspect in their paper.
- The research questions in section 4 seem to be appearing out of nowhere. They are relevant and tailored research questions; thus, they are valuable. However, they should be introduced earlier in the paper. Otherwise, they feel a bit detached from the rest of the paper.
- Why did the authors consider relying only on a single benchmarking dataset? The evaluation over a single dataset makes the obtained results unreliable. The authors must consider extending their evaluation to a dataset different from Defects4J to ensure the reliability and generalizability of the proposed approach.
- Similarly to the above, the authors tested their approach only over Java code bases. Thus, it is not clear whether the proposed framework could work on different programming languages. The authors must consider extending their evaluation to non-Java-based programs to ensure the generalizability of LLM4FL.
- Given that Defects4J was published in 2014 and might be quite outdated now, I’d suggest that the authors test LLM4FL on more recent benchmarks.
- The results presented in Table 3 are quite confusing and unreliable. What is the total number of faults? Why is the number of detected faults given as an absolute number rather than as a percentage over the total number of faults (675)? Also, the percentage gain is computed in a deceiving and conceptually wrong manner. It should be computed as the difference in percentage accuracy. Therefore, LLM4FL should have a 326/675 - 311/675 = 2.2 % improvement, while it should have 326/675 - 359/675 = - 4.89 % performance loss against DepGraph.
- LLM4FL is not the best approach out of the ones compared in Table 3. DepGraph outperforms LLM4FL across all settings. Therefore, I don’t understand how the authors can claim that LLM4FL is the new state-of-the-art.


[1]. Ciatto, Giovanni, et al. "Large language models as oracles for instantiating ontologies with domain-specific knowledge." Knowledge-Based Systems 310 (2025): 112940.

[2]. Dagan, Gautier, Frank Keller, and Alex Lascarides. "Dynamic planning with a llm." arXiv preprint arXiv:2308.06391 (2023).

[3]. Li, Haohang, et al. "Investorbench: A benchmark for financial decision-making tasks with llm-based agent." arXiv preprint arXiv:2412.18174 (2024).

[4]. Agiollo, Andrea, et al. "From large language models to small logic programs: building global explanations from disagreeing local post-hoc explainers." Autonomous Agents and Multi-Agent Systems 38.2 (2024): 32.

[5]. Barkan, Oren, et al. "LLM Explainability via Attributive Masking Learning." Findings of the Association for Computational Linguistics: EMNLP 2024. 2024.

[6]. Heyen, Henning, et al. "The effect of model size on llm post-hoc explainability via lime." arXiv preprint arXiv:2405.05348 (2024).

[7]. Bardhi, Enkeleda, et al. "Security and privacy of IP-ICN coexistence: A comprehensive survey." IEEE Communications Surveys & Tutorials 25.4 (2023): 2427-2455.

[8]. Park, Daeseung, et al. "A study on performance improvement of prompt engineering for generative AI with a large language model." Journal of web engineering 22.8 (2023): 1187-1206.

---

> ### Author Response · Authors · 2025-09-30
> **Clarifications on Generalizability, DepGraph Comparison, Evaluation Metrics, Agent Design, and SBFL Definition**
>
> Thank you for your detailed feedback. Below, we provide our responses to the main points raised.
>
> # Generalizability
> We focused on Java and Defects4J to address the language imbalance in recent studies on applying LLM for SE tasks: Cao et al. (2024) show that 95.6% of recent benchmarks are Python-only, including SWE-bench, HumanEval, MBPP, and RefactorBench. Java remains one of the most widely used languages in enterprise and open-source systems (TIOBE Software BV, 2025; Stack Overflow, 2024), and its structured grammar supports precise localization.
>
> # Comparison with DepGraph (State-of-the-art)
> LLM4FL does not outperform DepGraph, which achieves higher Top-1 accuracy. However, DepGraph is a supervised technique that requires extensive training data and significant computational resources, while LLM4FL operates in a zero-shot setting without task-specific training. Moreover, when we combine LLM4FL with DepGraph ordering, the combined approach surpasses when using DepGraph alone, highlighting that LLM4FL complements, rather than replaces, non-LLM and supervised ML techniques. Our contribution is not to claim state-of-the-art across all techniques, but rather to introduce a competitive multi-agent framework that narrows the gap with and complements supervised methods.
>
> # Evaluation Results Table 3
> We thank the reviewer for pointing out the confusing presentation in Table 3. We acknowledge that our current version reports absolute counts of detected faults along with relative improvements, which may misrepresent the actual performance differences. In the revised manuscript, we will (i) explicitly state the total number of faults (675), (ii) present accuracy values as percentages (e.g., 326/675 = 48.3%), and (iii) report performance differences in terms of percentage points rather than relative ratios. This change will make the results clearer and ensure that performance comparisons (e.g., LLM4FL vs. DepGraph) are transparent and easy to interpret.
>
> # Why three agents? Why these components?
> We designed the three-agent pipeline to address distinct challenges: (i) token limits (Context), (ii) repository-level navigation (Debugger), and (iii) stability of ranking (Reviewer). Each agent isolates and mitigates a known limitation of prior work. This mirrors human debugging, where developers first triage, then investigate dependencies, then review/refine their diagnosis.
>
> # Final “chain-of-thought” re-ranking necessity
> The chain-of-thought step is not a separate add-on but rather a component of the Reflexion process. Reflexion iteratively improves rankings within each group, and the final CoT step consolidates these refinements into a consistent global ranking across groups. We will clarify this more explicitly in the paper, as our ablation (Table 4) shows that removing the CoT from Reflexion reduces Top-1 accuracy by about 11%.
>
> # Spectrum-Based Fault Localization (SBFL)
> We thank the reviewer for pointing this out. SBFL (Spectrum-Based Fault Localization) formulas are statistical measures that compute a suspiciousness score for each program element (e.g., method) based on how often it is executed by failing vs. passing tests. The intuition is that: the more a method is executed by failing tests and the less it is executed by passing tests, the more suspicious it becomes. In the revision, we will briefly define SBFL in the introduction and give a simple example (e.g., the Ochiai formula).
>
> # References:
> * Cao, Jialun, et al. "How Should We Build A Benchmark? Revisiting 274 Code-Related Benchmarks For LLMs"
> * TIOBE Software. TIOBE Index. TIOBE, www.tiobe.com/tiobe-index/
> * Stack Overflow. 2024 Stack Overflow Developer Survey. Stack Overflow, survey.stackoverflow.co/2024/.

---

### Review · Reviewer_zJS6 · 2025-09-16

**Summary Of Contributions:**

The paper proposes an agentic framework for fault localization. The technique employs 3 agents that iteratively improve the ranking of fault containing candidates : (1) the Context Extraction Agent, (2) the Debugger Agent and, (3) the Reviewer Agent. Each agent operates on a set of candidates that contain faults.

(1) Code Extractor Agent obtains an initial ordering of candidates (at the method level) using GZoltar that makes use of spectrum-based fault localization. To account for context length of LLMs the agent divides the entire project in K groups, where each group at maximum has token length equal to the max token length of the LLM. Further the agent also does an LLM call to produce a summary of the stack trace, which is then used towards ranking methods in each group. The agent then performs a union over the reordered candidates to produce a “prioritized list of candidates”.

(2) The Debugger Agent obtains the prioritized list of candidates from the Context extractor agent and then performs call graph analysis to get a directed graph. The Debugger agent then analyzes the candidates using the “caller callee” relations to produce a set of failure reasons. In this step, each method is assigned an ordinal rank. The list of candidates from (1) is further refined using the ordinal rank.

(3) The Reviewer Agent finally performs reflexion to refine the list further till there is no improvement in ranking on the list from (2). It performs a “final” CoT step, where each agent generates a candidate fix for each method to reorder the reflexion generated candidate list.

Strengths:
- The paper empirically shows that LLM4FL outperforms other agentic approaches like AutoFL and SoapFL. It also outperforms DeepFL, a learning based approach
- The paper performs ablations showing the performance gains from using each agent

Weaknesses:
- The paper does not compare against AutoCodeRover, while mentioning it as an important piece of related work. Further, other agentic approaches like SWE-agent have also not been mentioned
- Grace (top-5 and top-10) and DepFL outperform LLM4FL. The paper lacks discussion on the performance gap, and analysis of the category of problems not addressed LLM4FL
- The method shows large swings ~22% in top-1 based on the initial ordering, suggesting results may be brittle to preprocessing choices (that are orthogonal to the core algorithm)
- Given that there are multiple design choices within each agent, the paper lacks ablations on sub choices:
   - In the Code extractor agent: How does granularity of the selected groups affect ranking/downstream performance?
   - In the Debugger Agent : How does an ordinal ranking strategy compare to score based ranking ?
   - In the reviewer Agent: What happens when you don’t use the final CoT step ?
- Generalizability of the approach: why wasn’t LLM4FL approach tried out on SWE-bench ?
- Generalizability of approach to other models (both open and closed source)
- The authors claim in Table 1 that other techniques are prone to hallucinations but do not substantiate it with any analysis (or justify how the graphRAG aids in reducing hallucinations)
- Another concern is the no-division ablation -- how many projects were out of the token limit ? how was this handled?
- The paper mentions stack traces might be long and verbose, hence argues to use summaries for Context Extraction Agent but then in the Debugger and Reviewer agents uses Stack Traces

**Additional Comments:**

Please address the missing citation on line 4 on page 11 - "_recent analysis confirms that Defects4J poses minimal data leakage risk for LLMs..._"

**Audience:**

Yes

**Audience Explanation:**

The paper studies an interesting issue. The work would benefit from another round of refinement before TMLR’s audience would find it compelling. In particular:
- Comparative gap: Comparison with AutoCodeRover (and adjacent agent systems like SWE-agent/mini-SWE-agent) and no SWE-bench evaluation, limiting relevance beyond Defects4J.
- Evidence gaps: Claims like hallucination reduction via Graph-RAG lack error analyses and significance tests.
- Scope & generality: Currently LLM4FL is restricted to Java-only projects, and no cross-model or cross-language sensitivity.
- Brittleness & ablations: Large ordering sensitivity (~22%) without mitigation, and missing ablations on key design choices (ranking scheme, group granularity, final CoT necessity).

**Claims And Evidence:**

No

**Claims Explanation:**

The following claims are well supported:
- Performance vs baselines - The paper uses the same underlying language model gpt-4o-mini for all the LLM based agentic approaches (AutoFL, AgentFL, SoapFL)
- The ablations show the importance for each agent at a high level.

While the above claims are well supported there are a number of claims that require more evidence:
- Comparison with AutoCodeRover (relevant Agentic pipeline that makes use of SBFL)
- The authors say that LLM4FL is the cheapest in terms of cost, but do not analyze other cost factors like latency, the number of LLM calls.
- Hallucination Reduction: The paper mentions that hallucinations are reduced when using GraphRAG but does not provide any evidence to support it
- Generality to other LLMs - The paper only studies LLM4FL in the context of gpt-4-mini.
- Generality to other projects defined in other programming languages.
- Use of stack traces: the context extractor agent makes use of summaries while the other two agents make use of stack traces

**Requested Changes:**

- Comparison with AutoCodeRover and SWE-agent
- Quantification of the hallucination claim against other techniques/baselines
- Addition of details on stack trace use
- Incorporating details like latency, number of calls, in the cost analysis section
- Analysis of sensitivity to initial ordering using different ordering techniques like Tarantula or D*
- Addition for more LLMs to quantify the robustness of the technique to the underlying LLM choice
- Experiments on datasets like SWE-bench, that show the generalizability of the approach
- Analysis of failure mode with actionable takeaways

---

> ### Author Response · Authors · 2025-09-30
> **Clarifications on Baselines, Design Choices, Ordering Sensitivity, and Generalizability of LLM4FL**
>
> Thank you for your detailed feedback. Below, we provide our responses to the main points raised.
>
> # Missing comparisons with key baselines
> While AutoCodeRover and SWE-agent are important related approaches, they rely on textual bug reports as the main input in the SWE-bench setting. The goal is to, given a bug report written in natural language, find the buggy location in the source code. In contrast, LLM4FL targets the scenario where tests fail and the goal is to locate the faulty method using information from the failing tests. There is no natural language description of the failure; only the test failure message, the corresponding stack traces, and test coverage are provided. Direct comparisons are not straightforward since one uses textual bug reports as input, while the other relies on failing tests, stack traces, and coverage data. To ensure fairness within our scenario, we evaluated LLM4FL against baselines that span all three major families of test-driven FL techniques: statistical (SBFL/Ochiai), learning-based (DeepFL, Grace, DepGraph), and LLM-based (AutoFL, SoapFL).
>
> # Performance gap vs Grace/DepGraph
> Grace and DepGraph outperform LLM4FL on some Top-N metrics, but these are supervised techniques that require training setup and the GNN architecture, which are high in cost and complexity. LLM4FL, in contrast, is a zero-shot approach requiring no task-specific training, making it more easily deployable. Moreover, when we combine LLM4FL with DepGraph ordering, the combined approach surpasses when using DepGraph alone, highlighting that LLM4FL complements, rather than replaces, non-LLM and supervised ML techniques.
>
> # Why three agents? Why these components?
> We designed the three-agent pipeline to address: (i) token limits (Context), (ii) repository-level navigation (Debugger), and (iii) stability of ranking (Reviewer). Each agent isolates and mitigates a known limitation of prior work. This mirrors human debugging, where developers first triage, then investigate dependencies, then review/refine their diagnosis.
>
> # Ordering sensitivity (~22% swings)
> We appreciate the observation regarding ordering sensitivity. Rather than indicating brittleness, we see this as an important empirical insight into how LLMs behave in fault localization. Our results show that different input orderings can change Top-1 accuracy by up to 22%, underscoring that ordering is a significant factor for LLM-based FL. This finding suggests that integrating traditional lightweight statistical ranking strategies (e.g., Ochiai) with LLM reasoning can be highly beneficial, and it opens a promising new line of research for the community.
>
> # Generalizability
> We focused on Java and Defects4J to address the language imbalance in SE benchmarks: Cao et al. (2024) show that 95.6% of recent benchmarks are Python-only, including SWE-bench, HumanEval, MBPP, and RefactorBench. Java remains one of the most widely used languages in enterprise and open-source systems (TIOBE Software BV, 2025; Stack Overflow, 2024), and its structured grammar supports precise localization. For models, we relied on lightweight GPT-4o-mini, which is popular, and applied it consistently across all LLM-based baselines. We chose GPT-4o-mini due to funding restrictions and because it is a powerful LLM, despite its smaller size. Our study is thus approach-centric rather than model-centric, and the framework can integrate any sufficiently capable LLM.
>
> # Stack trace inconsistency
> Sorry for the misunderstanding. All agents use pruned stack traces, but at different levels of detail. The Context Agent generates concise summaries (failure reason) to reduce verbosity, while the Debugger and Reviewer Agents use these failure reasons derived from the pruned stack-traces. This way, the level of detail is adjusted to each agent’s role, while keeping the overall approach consistent.
>
> # Design choice and ablations on sub choices
> Our ablations cover the three main components: removing Code Navigation (−16.5% Top-1), Division (−23.2%), and Reflexion (−11.3%). For the Debugger Agent, we used ordinal ranking since LLMs are more reliable at pairwise comparisons (“A is more suspicious than B”) than at numeric scoring, consistent with Liusie et al. (2024). For the Reviewer Agent, the chain-of-thought step is integral to Reflexion: it consolidates group-level refinements into a consistent global ranking, and removing it lowers Top-1 by ~11%.
>
> # References:
> * Cao, et al. "How Should We Build A Benchmark? Revisiting 274 Code-Related Benchmarks For LLMs"
> * TIOBE Software. TIOBE Index. TIOBE, www.tiobe.com/tiobe-index/
> * Stack Overflow. 2024 Stack Overflow Developer Survey. Stack Overflow, survey.stackoverflow.co/2024/.
> * Liusie et al. "LLM comparative assessment: Zero-shot NLG evaluation through pairwise comparisons using large language models.".

---

> > ### Comment · Reviewer_zJS6 · 2025-10-16
> >
> > Thanks for the response. I still have concerns about the following points:
> >
> > ## Missing comparisons with baselines
> >
> > SWE-agent is an agentic framework that supports various inputs. SWE-bench "the benchmark" has the property of providing issue text. Furthermore, recent work [1] has also explored using the SWE-agent for tasks such as compiler and logic repair. I believe that the authors can study SWE-agent and AutoCodeRover in a similar context while providing high-level instructions to locate a bug. Further comparison with more recent closed-source tools like Codex and Claude Code would also be appreciated (given that they do traversal on projects and are quite adept at fault localization).
> >
> >
> > ## Generalizability
> > I do not quite understand why "only" Java was considered towards this approach; there are other languages that might support precise localization, like C, Rust, C++, JS -- is there a challenge in adapting this approach towards other languages?
> >
> > ## Cost
> > While I understand there might be budget limitations, would it be possible to show results for open-source models?
> >
> > ## Points that remain unaddressed:
> > - The authors say that LLM4FL is the cheapest in terms of cost, but do not analyze other cost factors like latency, the number of LLM calls.
> > - Hallucination Reduction: The paper mentions that hallucinations are reduced when using GraphRAG but does not provide any evidence to support it
> > - Analysis of failure mode with actionable takeaways
> >
> > #### References:
> >
> > [1] Khatry et. al. CRUST-Bench: A Comprehensive Benchmark for C-to-safe-Rust Transpilation, COLM 2025

---

### Decision · Action_Editor_car9 · 2025-10-28

**Recommendation:** Reject

**Additional Comments:**

I think there are two approaches the authors can take. (1) Narrow the scope of the claims considerably: don't claim superior performance and scope the claims to just Defects4J.  (2) Broaden the evaluation of the method in terms of datasets and LLMs and strengthen the comparisons to other methods. This will likely lead to more long-term impact for the work.

**Audience:**

Yes

**Audience Explanation:**

Two of the reviewers indicated that they did not think this paper would be of interest to TMLR's audience. However, the prevalence of software engineering research in the area of LLMs for code makes me disagree. I think at least some readers could find this interesting, although the issues with the claims make it unsuitable for acceptance in its current state.

**Claims And Evidence:**

No

**Claims Explanation:**

This paper proposes a multi-agent approach to software fault localization, using a combination of three agents: Context Extraction, Debugger, and Reviewer.  These leverage a combination of LLMs and code analysis tools to identify and repair software faults. Evaluation on Defects4J shows performance stronger than some baselines but lower than SOTA specialized methods.

The main claims are (abbreviated and paraphrased by me):

1. Introduction of a novel method

2. Evaluation demonstrating "superior performance"

3. Some conclusions from analysis and investigation

Reviewers generally found a few aspects of this work to be lacking, related to claims 1 and 2. First, the evaluation on just Defects4J does not substantiate the broader claims about the utility of this method.  Second, the system presents an ad hoc collection of modules, and does not sufficiently evaluate the design choices of the presented system.  Although novel, it is hard to have a strong scientific takeaway from the content of this paper. Finally, the performance is lower than DepGraph without sufficient justification; the authors present some claims that the present approach can work with "insufficient training data" and that training DepGraph is expensive, but I do not see these as strong arguments. Some other potential comparison points and baselines are missed as well.

Therefore, the reviewers and I do not consider the claims to be sufficiently supported.

**Resubmission Of Major Revision:**

The authors may consider submitting a major revision at a later time.